# Biodiversity of *rolB/C*-like Natural Transgene in the Genus *Vaccinium* L. and Its Application for Phylogenetic Studies

**DOI:** 10.3390/ijms24086932

**Published:** 2023-04-08

**Authors:** Roman Zhidkin, Peter Zhurbenko, Olesya Bogomaz, Elizaveta Gorodilova, Ivan Katsapov, Dmitry Antropov, Tatiana Matveeva

**Affiliations:** 1Department of Genetic and Breeding, St. Petersburg State University, Saint Petersburg 199034, Russia; st085586@student.spbu.ru (R.Z.);; 2Komarov Botanical Institute of the Russian Academy of Sciences, Saint Petersburg 197022, Russia; 3Faculty of Bioengineering and Bioinformatics, Moscow State University, Moscow 119991, Russia; 4Research Park, St. Petersburg State University, Saint Petersburg 199034, Russia

**Keywords:** *Vaccinium* L., cellular T-DNA, allele phasing, phylogenetics

## Abstract

A variety of plant species found in nature contain agrobacterial T-DNAs in their genomes which they transmit in a series of sexual generations. Such T-DNAs are called cellular T-DNAs (cT-DNAs). cT-DNAs have been discovered in dozens of plant genera, and are suggested to be used in phylogenetic studies, since they are well-defined and unrelated to other plant sequences. Their integration into a particular chromosomal site indicates a founder event and a clear start of a new clade. cT-DNA inserts do not disseminate in the genome after insertion. They can be large and old enough to generate a range of variants, thereby allowing the construction of detailed trees. Unusual cT-DNAs (containing the *rolB/C*-like gene) were found in our previous study in the genome data of two *Vaccinium* L. species. Here, we present a deeper study of these sequences in *Vaccinium* L. Molecular-genetic and bioinformatics methods were applied for sequencing, assembly, and analysis of the *rolB/C*-like gene. The *rolB/C*-like gene was discovered in 26 new *Vaccinium* species and *Agapetes serpens* (Wight) Sleumer. Most samples were found to contain full-size genes. It allowed us to develop approaches for the phasing of cT-DNA alleles and reconstruct a *Vaccinium* phylogenetic relationship. Intra- and interspecific polymorphism found in cT-DNA makes it possible to use it for phylogenetic and phylogeographic studies of the *Vaccinium* genus.

## 1. Introduction

“*Agrobacterium*” is a soil bacterium with a unique mechanism to transfer well-defined DNA fragments (T-DNAs) into the chromosomes of a large variety of dicotyledonous plant species [1,2,3,4,5,6]. Normally, the transfers result in crown gall disease (in the case of *Agrobacterium tumefaciens* Conn or *A. vitis* Ophel and Kerr) or hairy roots formation (in the case of *A. rhizogenes*, also called *Rhizobium rhizogenes* Young). It has been experimentally demonstrated that hairy roots can regenerate into fertile plants [7,8,9]. Moreover, a variety of plant species, found in nature, contain T-DNA in their genomes, which they transmit in a series of sexual generations. Such T-DNA is called cellular (cT-DNA), and the plants containing it are called naturally transgenic or genetically-modified organisms (nGMO). Initially, such sequences were discovered in *Nicotiana glauca* Graham and other *Nicotiana* L. species [10,11,12], and further nGMOs were found inside genera *Linaria* Mill. [13], *Ipomoea* L. [14], *Cuscuta* L. [15], and dozens of other species [6,16,17,18]. These discoveries enabled making a number of generalizations regarding the distribution, structures, and possible functions of cT-DNA, summarized in our previous paper [19]. Sequences of cT-DNAs have been used in phylogenetic studies of *Linaria* [20], *Ipomoea* [21], *Nicotiana* [22], and *Camellia* [23]. They present several advantages for exploring the origin and evolution of nGMO species. To begin with, cT-DNAs are well-defined, highly specific, and well recognizable DNA fragments, unrelated to plant sequences. They do not occur in non-transformed ancestors, and their integration at a particular chromosomal site constitutes a founder event which creates a clear start for a new clade. Moreover, unlike transposable elements, cT-DNA inserts do not amplify and disseminate in the genome after insertion. This prevents the generation of multiple paralogs, which is a significant advantage over classical nuclear markers. Additionally, cT-DNAs can be large and old enough to generate a range of variants, thereby allowing the construction of detailed trees [23]. Phasing of T-DNA alleles enables the reconstruction of more precise phylogenetic relationships and identification of hybridization events in the evolution of plant species [24]. These approaches have been successfully applied in the study of the genus *Camellia* [23].

We also found unusual sequences of the *rolB/C*-like gene in the genome data of *Vaccinium corymbosum* L. and *Vaccinium macrocarpon* Ait. [6,16]. This gene belongs to the *plast*-gene family. It is more similar to *plast*-genes present in fungus *Laccaria bicolor* (Maire) P.D.Orton (and absent in most basidiomycetes), plant *Nyssa sinensis* Oliv., and bacteria *Ensifer* sp. Casida, than to ones from known strains of *Agrobacterium* (*Rhizobium*) *rhizogenes*, which indicates the transfer of this sequence from an unknown «Agrobacterium» strain [16].

Based on the results of previous phylogenetic studies, we can try to reconstruct phylogenetic relationships between species of *Vaccinium* genus, using the *rolB/C*-like gene sequence. This task is quite relevant since the phylogeny of *Vaccinium* genus is highly controversial.

The *Vaccinium* genus was first described by Carl Linnaeus in 1753 [25]. After that, there were no large-scale phylogenetic studies of this genus until the beginning of the 20th century, when active breeding of cranberries and blueberries started in the United States [26,27]. Therefore, the first studies of *Vaccinium* phylogeny and taxonomy mainly involved species distributed in North America. The main difficulties in such studies were caused by the absence of fertility barriers in morphologically different organisms, which leads to the formation of a large number of hybrids, along with polyploidy distribution throughout the genus [27].

Camp was one of the first to introduce the system of *Vaccinium* genus in 1945 [28]. He conducted his classification according to morphological features. As a result, the genus was divided into several sections. The North American blueberry section *Cyanococcus* included 9 diploid, 12 tetraploid, and 3 hexaploid species [29]. Later, some of the species were merged into *V. corymbosum* with three levels of ploidy; therefore, in the new Kloet’s classification, the *Cyanococcus* section contained 6 diploid, 5 tetraploid, and 1 hexaploid blueberry species [29], while dividing into sections did not change itself [30].

Thereby, *Vaccinium* species differentiation is complicated by polyploidy, similar morphology, and introgressions during hybridization [29]. Additionally, the application of morphological traits for phylogenetic studies of this genus does not always enable an ambiguous assessment of the evolutionary relationships between the studied species. This problem can be partially solved by the application of modern methods of phylogeny.

Methods such as DNA barcoding, fingerprinting, phylogenomics, and chemosystematics have been used recently in the study of *Vaccinium* genus. They allowed us to reconsider the taxonomy proposed by K. Kloet. An example of such study is the application of matK and ITS markers to determine phylogenetic relationships of various representatives of the entire tribe Vaccinieae [31]. The resulting dendrograms did not confirm traditional generic boundaries, but several well-supported clades were found on the tree: Andean; Mesoamerican/Caribbean; East Malaysian; Agapetes (consisting of some Asiatic *Vaccinium* and *Agapetes)*; Bracteata-Oarianthe (represented by the respective sections); Orthaea/Notopora (containing the genera *Orthaea* Klotzsch and *Notopora* Hook); Myrtillus; and Vaccinium clades. Moreover, most of the discovered clades combined representatives of various genera, while the clades Vaccinium and Myrtillus contained species from different *Vaccinium* sections according to the previous classifications. Based on the obtained results, the authors concluded that it is necessary to reassess the taxonomy of the genus *Vaccinium*, since it is not monophyletic. Although work on this reassessment began in 2003 [32], a number of researchers pointed at too radical differences from the accepted structure of the genus. Inconsistencies can be explained by the difficult interpretation of the results of phylogenetic analysis based on ITS and *matK*, in taxon, where hybridization and polyploidization played a significant evolutionary role [33].

Since the use of DNA barcoding does not enable unambiguous reconstruction of the phylogeny of *Vaccinium* genus, more time-consuming and expensive methods of molecular phylogenetics were used in genetic studies of economically important species. Recent assemblies of the genomes of *V. macrocarpon*, *V. microcarpum* (Turcz. ex Rupr.) Schmalh., *V. oxycoccos* L., and *V. corymbosum* made it possible to conduct their comparative genomics [34,35]. Molecular dating has shown that *V. macrocarpon* diverged from *V. oxycoccos* 2 mya, and from *V. microcarpum* 4.5 mya. Divergence time estimated for *V. macrocarpon* and *V. corymbosum* differs from 5 to 10.4 mya [34,35].

An analysis of the intra- and interspecific variability of American blueberry species by double digest restriction-site-associated DNA sequencing [36], showed that the rabbit eye blueberry and northern highbush blueberry are relatively homogeneous, but the southern highbush blueberry contains a significantly more mixed genetic background. Considering the pedigree of blueberries, the most optimal solution was to separate the entire dataset into nine hypothetical genomes, which correspond to the number of species actively used in breeding, *V. darrowii* Camp, *V. elliottii* Chapm, *V. tenellum* Ait., *V. angustifolium* Ait., *V. corymbosum*, *V. constablaei* A. Gray, *V. virgatum* Ait, *V. myrtilloides* Michx., and *V. pallidum* Ait. The trends identified in this way are consistent with the history of blueberry breeding [36].

Thereby, attempts to reconstruct the phylogenetic relationships of species of the genus *Vaccinium* by various methods have led to contradictory results. The use of phylogenetic markers ITS and matK leads to the construction of a phylogenetic tree with significant differences from the classical system. The use of more expensive and time-consuming methods of genome studies gives a more plausible picture, that is more consistent with the classical system. However, expensive methods can currently be applied to a narrow list of economically important species. For more extensive studies of various representatives of the genus, easy-to-use and cheap markers are required. They can be developed based on genome regions newly acquired in the course of evolution with a known structure and localization. An example of such sequences would be cellular T-DNA. Therefore, our study aims to characterize cellular T-DNA polymorphism in the genomes of plants of *Vaccinium* genus, and to evaluate the possibility of its application for phylogenetic studies of the genus.

## 2. Results

### 2.1. Vaccinium rolB/C-like Gene: General Description

Until recently, the *rolB/C*-like gene has been described only in *V. corymbosum* and *V. macrocarpon*. In the framework of this study, using BLAST algorithm, this gene was described in the WGS data for *V. myrtillus* and *V. darrowii*. Their ORFs correspond to the amino acid sequences from 290 (in *V. myrtillus*) to 292 amino acids (in *V. macrocarpon*).

In blueberries, the gene is located on chromosome 3 (in the cranberry genome, this region corresponds to chromosome 4). Located on both sides of it, at a distance of 5 kb, are genes of plant origin with functions that have not been described yet. No other agrobacterial genes were found in the genome of plants of *Vaccinium* genus.

Our search for homologues of the *rolB/C*-like gene in SRA database revealed new species of naturally transgenic plants in the *Vaccinium* genus, summarized in Table 1.

Natural GMOs were found in the sections *Oxycoccoides*, *Oxycoccus*, *Cyanococcus*, *Hemimyrtillus*, *Myrtillus*, *Polycodium*, *Bracteata*, *Vaccinium*, and *Vitis-idaea*, indicating that their common ancestor was transformed before the sections diverged.

Genomic data for different species varies in coverage quality. In some cases, the presence of a transgene can be inferred from single reads. In other cases (in *V. selerianum*, *V. shastense*, *V. striicaule*, *V. symplocifolium*, *V. vidalii*, and *V. whitmeei*), the transgene was not found at all, but this does not mean its absence from the genome, it indicates, however, that additional studies are required.

Sequences with good coverage were used further to assemble full-length sequences. For a deeper understanding of the gene variability, field material was collected, which was included in further variability analysis of the *rolB/C*-like gene. In all studied species, except for *V*. *oxycoccos* (see below)*,* only full-length sequences of the *roB/C*-like gene were found.

When studying SNPs within full-length sequences, one can see that there are nucleotide differences characteristic of the species, distinguishing each of them from other representatives of the genus *Vaccinium*. They are represented by substitutions and indels. Moreover, indels are equal or multiples of three nucleotides. This is evidence in favor of the gene product functionality. Examples of such polymorphism can be SNPs in *V. uliginosum* and *V. vitis-idaea* shown in Appendix A, as well as indels in different *Vaccinium* species shown in Appendix A.

Modeling of protein structures based on the predicted amino acid sequences (Appendix A) shows their similarity in representatives of the genus *Vaccinium*, as well as in *Laccaria*, *Nyssa*, and *Ensifer*. The closest of the described agrobacterial proteins is the RolB protein (Figure 1).

Collectively, these data allow us to outline the directions for future research in terms of these sequences’ functionality.

### 2.2. Intraspecific Variability of rolB/C-like Gene

To assess intraspecific variability of the *rolB/C*-like gene, we used sequences of several independent samples presented in the SRA for *V. bracteatum*, *V. corymbosum*, *V. darrowii*, *V. dunalianum*, *V. macrocarpon*, *V. myrtillus*, *V. oxycoccos*, *V. uliginosum*, *V. virgatum*/*ashei*, and *V. vitis-idaea*. In addition, samples of *V. myrtillus*, *V. oxycoccos*, *V. uliginosum*, and *V. vitis-idaea* collected from different geographic locations were characterized for polymorphism of the studied gene. For each genotype, alleles were reconstructed, based on Sanger sequencing data, or SRA reads.

PCR with primers VaccR and VaccF, followed by sequencing of the products have shown that 8 out of 15 samples of *V. oxycoccos* were homozygous for a deletion spanning from nucleotides 198 to 800 of CDS (Figure 2). Three samples had deletions other than the most common. They spanned nucleotides 354–798, 390–548, and 159 to the end of the gene, respectively. Four samples contained full size genes. All samples of other species contained only full-size intact genes.

Within each species, there are SNPs represented by nucleotide substitutions that distinguish different alleles of a gene from each other. Based on these data, we can assess the diversity of alleles within a species, the frequency of occurrence of homo- and heterozygotes in populations, and the geographical distribution of alleles for some species. Data on the number of investigated alleles are presented in Table 2. Figure 3 shows allele frequency diagrams for *Vaccinium* species. This diagram does not claim to be an accurate estimate of allele frequencies, however, it stimulates the consideration that there are more common and rarer alleles. For example, A in *V. vitis-idaea* was found in most of the analyzed samples. The remaining alleles were found in a smaller number of samples and sometimes specific for certain locations. The proportion of homo- and heterozygotes and allelic diversity differ in the studied species, which can be explained by different reproduction biology. For example, blueberry species are more prone to cross-pollination, compared to cranberries.

Reconstruction of the phylogenetic relationships, based on molecular genetic studies of taxonomically significant regions of DNA, often relies on the evaluation of single specimens from the studied species. To avoid possible mistakes, especially when working with new markers, it is logical to test them out on a wider material. Since we used samples from collection points that are geographically distant from each other, it was decided to compare interspecific and intraspecific differences between them. Independent alleles were used for phylogenetic tree construction. Only full-length sequences were included in the analysis (Figure 4).

The resulting phylogenetic tree, in general, was consistent with our expectations. A total of 8 clades, that mostly correspond to analyzed species, can be distinguished on this tree. Interestingly, the clades represented by wild species (A, B, E, F) are homogeneous, despite the wide geography of sampling. At the same time, clades D and H are mosaic. Clade H is represented by alleles from *V. virgatum* and *V. ashei* Reade. Species name given by the authors of sequences was kept, but at the same time, we understand them as synonyms and refer them to the same species. Two *V. corymbosym* alleles C and E are noted in this clade. Both of them belong to the variety Draper, in the breeding of which *V. virgatum* and *V. darrowii* were involved [37]. Clade G consists of the related species *V. corymbosum* and *V. darrowii*, with an admixture of two alleles of *V. virgatum*. Their presence may be explained by hybridization during development of cultivars of North American Rabbit-eye blueberries.

*V. macrocarpon* and *V. oxycoccos* are related species that form clade D. These species require more detailed study in the future, since the American cranberry is a cultivated species, and, therefore, hybridization was used in the early stages of its breeding. In turn, swamp cranberries are the only species known today where the transgene was extensively mutated in many of the studied samples.

In general, our studies have shown that, at least in the analyzed species, the intraspecific variability of transgene is lower than interspecific, and the mosaic distribution of alleles on the tree in American blueberries can be explained in terms of interspecific hybridization documented during the creation of studied varieties.

Therefore, we compiled consensus sequences, as a characteristic of each of the species analyzed at the previous stage, and supplemented this list with sequences obtained on the basis of gene sequences we obtained from single samples of the collection of the Komarov Botanical Institute.

### 2.3. Reconstruction of Phylogenetic Relationship of Vaccinium Species Based on rolB/C-like Sequence

In the final part of this study, several types of data were used. First of which, were the consensus sequences of the *rolB/C*-like gene of species studied in the previous stage. Alleles of presumably hybrid origin were excluded at the stage of constructing consensus sequences.

In addition, sequences of species sufficient for gene assembly but not sufficient for allele phasing were included in our analysis.

Finally, we obtained sequences of the *rolB/C*-like gene from additional species, represented by single samples in the collection of the Komarov Botanical Institute.

The phylogenetic tree presented in Figure 5 included, in addition to studied above, such species as *V. myrtilloides* (Sec. *Cyanococcus*), *V. ovalifolium* Sm. (Sec. *Myrtillus*), *V. praestans* Lamb. (Sec. *Praestantia*), *V. smallii* A. Gray (Sec. *Hemimyrtillus*), *V. vulcanorum* Kom. (Sec. *Vaccinium*), *V. conchophyllum* Rehder, *V. emarginatum* Hayata (Sec. *Conchophyllum*), and *Agapetes serpens* (Wight) Sleumer (based on previous studies, *Agapetes* can be considered a closely related genus in relation to *Vaccinium*, so its involvement in the work seems logical) [31].

As can be seen from Figure 5, species assigned to the same section in the classical system of *Vaccinium* genus are clustered together when constructing a tree based on the proposed DNA marker. Marked with ovals in Figure 5.

Another phylogenetic tree was constructed based on ITS sequences of samples studied in this work, combined with sequences from Kron et al. research [31]. This tree is presented in Figure 6. We can distinguish Andean, Meso-American/Caribbean, East Malaysian, Agapetes, Bracteata-Oarianthe, Orthaea/Notopora, Myrtillus, and Vaccinium clades, which were previously identified by Kron et al., composition of these clades differs significantly from the traditional taxonomy. Thereby, species *V. macrocarpon* and *V. oxycoccos* belong to the section *Oxycoccus*, but on ITS tree these species are separated by species *V. vitis-idaea*, which belongs to the section *Vitis-idaea*, Clade Myrtillus unites species of sections *Myrtillus* and *Macropelma*. Clade Bracteata-Oarianthe also includes species from sections *Hemimyrtillus*, *Oarianthe*, *Bracteata*, and *Baccula-Nigra*. Clade Agapetes includes representatives of genus *Agapetes* and some representatives of genus *Vaccinium* from sections *Rigiolepis*, *Conchophyllum*, and *Galeopetalum*, while some other species from these sections are spread over the tree. Orthaea/Notopora clade includes representatives of different genera, *Notopora*, *Orthaea*, and *Vaccinium*. The combination of *Vaccinium* representatives with other genera is also characteristic of Meso-American/Caribbean clades, which includes a representative of section *Oreades*.

There are much fewer of such contradictions in the phylogenetic tree based on the *rolB/C*-like gene.

## 3. Discussion

*Vaccinium* is the fifth genus of natural GMOs after *Nicotiana*, *Linaria*, *Ipomoea,* and *Camellia*, where the cT-DNA structure was characterized in greater detail in many species. [11,13,14,23].

In addition to the unusual *plast*-gene found in plant genomes, a distinguishing feature of the *Vaccinium* genus is also the fact that the gene has been preserved intact in most of the studied species. The *plast*-genes family includes most of the oncogenes of *R. rhizogenes* (coding for RolB, RolC, Orf13, and Orf14 proteins), and some T-DNA genes of *A. tumefaciens* (coding for p5, p7, 6a, 6b) [38]. They have also been found in naturally transgenic plants, fungi, and some other bacteria [10,13,14,38]. *Plast* genes were described in detail in the review by Leon Otten [38], where he defined them on the basis of weak, but significant, protein similarity, and highlighted their capacity to change plant development, being introduced into plant genomes during the genetic transformation process. Levesque et al. [39] suggested that *plast* genes could have similar functions because of their common ancestry, and their diversification could be an adaptation to different plant species. Describing the biodiversity of *plast* genes can be very important in order to find their basic function and some common features of all these genes. In this regard, the discovery of *plast* genes, that are not quite similar to previously known ones (as in our case), is very valuable for future research.

In accordance with the analysis of previously assembled genomes and sequences, the representatives of two *Vaccinium* subgenera and *Agapetes* have a common localization site of the *rolB/C*-like sequence in the genome. This indicates a common origin of the sequence, as a result of a single transformation event of the ancestral form, and, accordingly, the monophyletic origin of all species containing transgene, included in our analysis. During evolution, the newly acquired sequence seems to have accumulated mutations. This variability can be assessed and used to estimate species divergence.

This is very important for the *Vaccinium* genus since there is still no consensus on the genus system.

Phylogenetic studies of the *Vaccinium* genus have been conducted since the middle of last century [26,27]. The traditional system of the genus involves its division into 2 subgenera and 33 sections. This division is based on morphological features [30]. The use of DNA barcoding methods showed the polyphyletic nature of the genus. At the same time, the construction of phylogenetic trees, based on ITS and *matK*, revealed contradictions with the traditional system. The trees constructed using ITS sequences by Kron et al. [31], and reconstructed in our study, based on an extended list of species, have shown a similar topology. They contained several well-maintained clades, such as Andean; Mesoamerican/Caribbean; East Malaysian; Agapetes, consisting of some Asiatic *Vaccinium* and *Agapetes*; Bracteata-Oarianthe, uniting representatives of the respective sections; Orthaea/Notopora, which includes the genera Orthaea and Notopora; *Myrtillus;* and *Vaccinium*. These clades contained species of the genus *Vaccinium*, which, in previous classifications, belonged to different sections. Moreover, representatives of the *oxycoccus* subgenus were separated on the tree by *V. vitis-idea*. Representatives of the *Cyanococcus* section also fell into different clades, clustering with species of other sections.

This contradiction could be related to the manifestation peculiarities of used DNA markers in species of hybrid origin. The ITS of hybrid species retain the features of only one parent, and chloroplast markers are inherited mainly through the maternal line [40]. At the same time, the phylogeny based on genomic data is in better agreement with the traditional system [34,35,41,42]. Collectively, these data indicate that additional research is required, including the search for new cheap and reliable markers for the molecular phylogeny of *Vaccinium*.

In this article, we propose such a marker. An important feature of the work carried out by us was the use of a large number of forms of cranberries, lingonberries, and blueberries of different geographical origin, as well as the phasing of natural transgene alleles using SRA data in order to assess intraspecific variability. Such an approach was first used to study cellular T-DNA of representatives of the *Camellia* genus [23]. The analysis of individual alleles in phylogenetic studies makes it possible to identify acts of hybridization, including interspecific hybridization. This is important to understand when assessing the degree of speciation. Alleles identification makes it possible to better describe intraspecific variability in order to assess the occurrence of alleles and their geography. In the future, this marker can be used to describe the allelic state of a gene in various populations in order to mark them and control unauthorized picking of wild berries. In addition to the general assessment of variability, the description of alleles makes it possible to identify the facts of hybridization in the pedigree of specific isolates, varieties, lines of cultivated *Vaccinium* plants, as well as to identify homo- and heterozygotes for this marker. This may be useful in interpreting controversial results of the origin of hybrid forms, as well as illustrating the incomplete isolation of species, as previously demonstrated in case of the genus *Camellia* [23].

Based on the phylogenetic analysis of allelic variants, obtained in our research, it can be concluded that intraspecific variability of marker in *Vaccinium* is less than the interspecific one. Therefore, it can be used to study the phylogenetic relationships of *Vaccinium* species.

A phylogenetic analysis of the genus *Vaccinium* was performed, using sequences of the *rolB/C*-like gene, as well as ITS sequences from previous studies [31], supplemented by sequences of our samples. The addition of new species to phylogenetic analysis, based on ITS, did not significantly change the topology of the tree.

The comparison of trees built based on ITS and *rolB/C*-like gene shows that they have similarities and differences. Species of *Myrtillus* section (*V. myrtillus* and *V. ovalifolium*) cluster together in both trees. Species of *Cyanococcus* section are clustered on both trees in a similar way. No other similarities in the topology of these trees were found. Species *V. conchophyllum* and *V. emarginatum* belonging to section *Conchophyllum* cluster together on the *rolB/C*-like tree, but not on ITS tree. Species *V. macrocarpon* and *V. oxycoccos* from section *Oxycoccus, V. vulcanorum* and *V. uliginosum* from section *Vaccinium* form separate clades on the *rolB/C*-like tree, but are scattered across different parts of the ITS tree.

*Bracteata*, *Hemimyrtillus*, *Vitis-idaea*, *Oxycoccoides*, and *Praestantia* sections are represented by single species on the *rolB/C*-tree. These sections require additional research involving new species. Reconstruction of phylogeny using the *rolB/C*-like gene is consistent with data obtained by other researchers using more expensive and time-consuming methods [34,35,42].

Therefore, a phylogeny, built on the basis of the *rolB/C*-like gene, has fewer contradictions with classical ideas, compared to a tree built based on ITS. The *rolB/C*-like gene, as a molecular marker, is cheap, easy to use, and is not time consuming. Given minimal amounts of biological material, new species can easily be included in further analyses.

## 4. Materials and Methods

### 4.1. Plant Material

Plant material was represented by samples of *Vaccinium myrtillus* L., *V. oxycoccos*, *V. vitis-idaea* L., and *V. uliginosum* L., collected from various habitats in Russia and Belgium. Samples of 9 species of the studied genus were taken from the collection of the Komarov Botanical Institute (St. Petersburg, Russia). A detailed description of the samples is presented in Appendix A.

### 4.2. SRA Data

Sequences of *Vaccinium* species from SRA database, used in this study are presented in Appendix A.

### 4.3. DNA Isolation

Leaves and stems of plants were grounded in liquid nitrogen. Plant DNA was isolated by a CTAB method described by Draper and co-authors [43].

### 4.4. PCR

Type I. Classic PCR was performed using the primers shown in Table 3, in the Tercyc gene cycler (DNA Technology, Moscow, Russia). Primers were selected on the basis of cranberry genome fragment (JOTO01169953.1). The 40 µL reaction mixture contained DreamTaq Green PCR Master Mix (2X) buffer (Thermo Fisher, Waltham, MA, USA), 100 ng DNA, and 10 µM primers. The following program was used for PCR, 5 min at 93 °C; 40 cycles of 17 s at 93 °C, 30 s at X °C and 90 s at 72 °C; then 5 min at 72 °C, where X was the annealing temperature, depending on the primers used.

Type II. Real-time PCR from colonies was performed for additional control of early threshold values to search for clones containing the target insert. Real-time PCR was carried out in ANK-32 cycler (Sintol, Moscow, Russia). Primers for plasmid pJET1.2 from the CloneJET PCR Cloning Kit (Thermo Fisher, Waltham, MA, USA) were used in the reaction. The 20 μL of the reaction mixture contained SsoAdvanced Universal SYBR Green Supermix buffer (Bio-Rad, Hercules, CA, USA), 10 μM primers, and a suspension of *E. coli* bacterial cells taken from a colony of transformants. The following program was used for real-time PCR: 60 s at 50 °C, 40 cycles of 5 min at 95 °C, 30 s at 58 °C, 60 s 72 °C, and 18 s at 95 °C.

PCR products were separated on agarose gel in 1x TBE buffer and visualized using GelDoc Go (BioRad, Hercules, CA, USA).

### 4.5. Molecular Cloning

PCR products were cloned into pJET1.2 vector, using CloneJET PCR Cloning Kit (Thermo Fisher, Waltham, MA, USA) according to the manufacturer instructions, and transformed into DH5 alpha chemically competent cells according to Inoue and co-authors [44].

### 4.6. DNA Sequencing

PCR fragments were sequenced using Sanger method, and BrilliantDye™ Terminator (v3.1) Cycle Sequencing Kit (NimaGen, Nijmegen, The Netherlands). Then, sequencing mixtures were separated at the Resource Center of St. Petersburg University “Development of Molecular and Cellular Technologies” using an ABI Prism 3500 xl sequencer (Applied Biosystems, Waltham, MA, USA).

### 4.7. Allele Reconstruction from Sanger Sequencing Data

In order to determine the sequences of alleles, we represented in binary form each sample as a set of polymorphic positions, where “1” indicates the most frequent nucleotide in a given position in the species, and “0” the least frequent. Then, for each sample, all possible combinations of values were written for polymorphic positions. Since homozygotes and samples with one SNP were found among the samples, we were able to obtain a primary pool of alleles, and these alleles were subsequently found in the remaining samples. Since each of these alleles, in a diploid, must match to an allele with alternative values in polymorphic positions, a homologous pair can be found for the primary allele [45].

In cases where the number of sequences was insufficient to resolve the set of sequences per allele, they were cloned into pJET 1.2 according to Clone Jet PCR Cloning Kit (Thermo Fisher Scientific, Waltham, MA, USA) manufacturer’s instructions. Then, inserts from individual clones, that represent individual alleles, were sequenced.

### 4.8. Allele Phasing

The NCBI SRA database was used to search for T-DNA in the *Vaccinium* genus. The search was carried out using BLAST against the reference (JOTO01169953.1). Resulting reads were aligned to the reference using BWA 0.7.17 [46]. The processing of SAM files was carried out in SAMtools 1.7 [47]. The alignment visualization for ploidy estimation was performed in IGV 2.12.3 [48]. Allele phasing was performed using variant calling with GATK 4.2 [49] followed by WhatsHap 1.0 [50]. Additionally, the SAMtools phase was used for alternative analysis. In the case of small coverage, alleles were phased manually, with some only partially successful. Upon the assumption of SNP being artificial, its presence was verified in other samples.

### 4.9. Phylogenetic Analysis

Sequences were aligned using the MAFFT online service for multiple sequence alignment [51]. The evolutionary history was inferred by using the Maximum Likelihood method and General Time Reversible model [52]. Initial tree(s) for the heuristic search were obtained automatically by applying Neighbor-Join and BioNJ algorithms to a matrix of pairwise distances estimated using the Maximum Composite Likelihood (MCL) approach, and then selecting the topology with superior log likelihood value. Trees were drawn to scale, with branch lengths measured in the number of substitutions per site. All positions with less than 95% site coverage were eliminated, i.e., fewer than 5% alignment gaps, missing data, and ambiguous bases were allowed at any position (partial deletion option). Evolutionary analyses were conducted in MEGA11 [53].

### 4.10. Prediction of Protein’s Structure

AlphaFold prediction [54] of the structures of various Plast-proteins was made using the Colabfold web resource at https://colab.research.google.com/github/sokrypton/ColabFold/blob/main/AlphaFold2.ipynb (accessed on 18 April 2022), using default settings.

## 5. Conclusions

In this study, new species of natural GMOs were described within the genera *Vaccinium* and *Agapetes*. The natural transgene of these plants can be used for phylogenetic studies of the genus. This nuclear marker is cheap, easy to use, and is not time consuming. It can also be recommended for further research. The allele phasing approach makes it possible to track hybridization events in the evolution of studied plants. Data, regarding intraspecific variability of the proposed marker, can be used to mark populations of wild berries of the *Vaccinium* genus, in order to prevent their unauthorized harvesting.

## Figures and Tables

**Figure 1 ijms-24-06932-f001:**
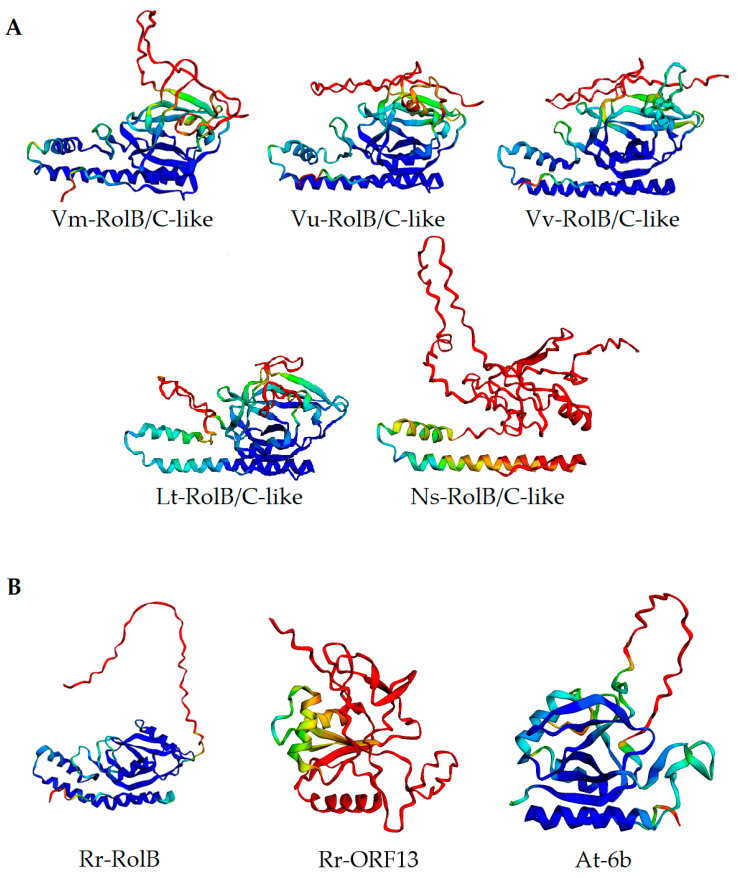
Predicted structures of various Plast-proteins. (**A**) RolB/C-protein from plants and fungus; (**B**) various plast-proteins from *Agrobacterium*. Abbreviations: Vm—*V. macrocarpon*, Vu—*V. uliginosum*, Vv—*V. vitis-idaea*, Lt—*L. trichodermophora*, Ns—*N. sinensis*, Rr—*R. rhizogenes*, and At—*A. tumefaciens*.

**Figure 2 ijms-24-06932-f002:**
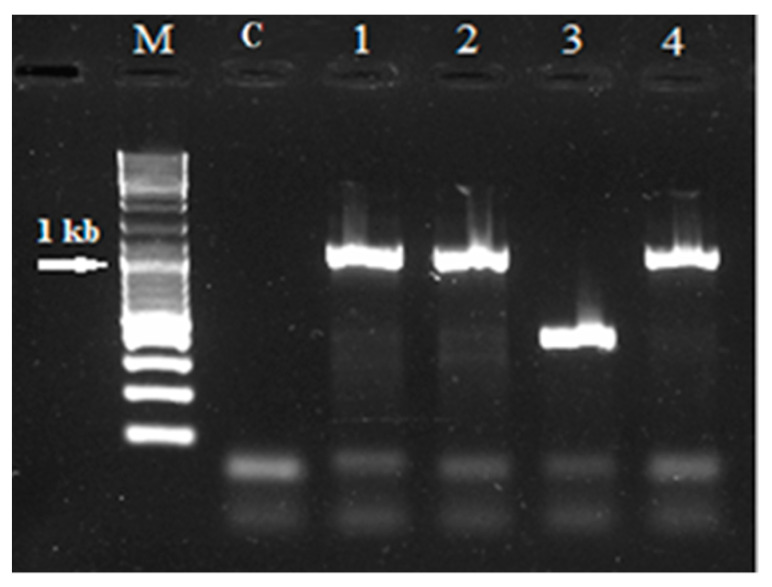
PCR products containing the *rolB/C*-like gene in cranberry samples. (M) Molecular Weight Marker GeneRuler^TM^ 1 kb DNA Ladder (Thermo Fisher Scientific, USA); (C) negative control; (1,2,3)—*V. oxycoccos* (samples 1,2—full size, 3—the most common deletion); and (4) *V. macrocarpon*.

**Figure 3 ijms-24-06932-f003:**
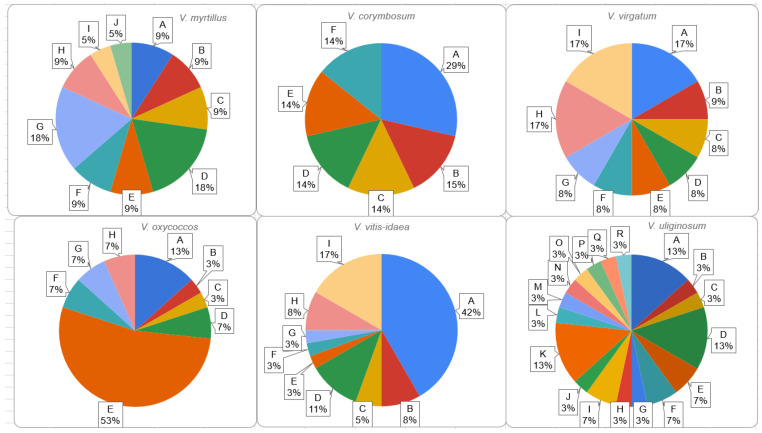
Frequencies of alleles of *rolB/C*-like gene in *Vaccinium* species.

**Figure 4 ijms-24-06932-f004:**
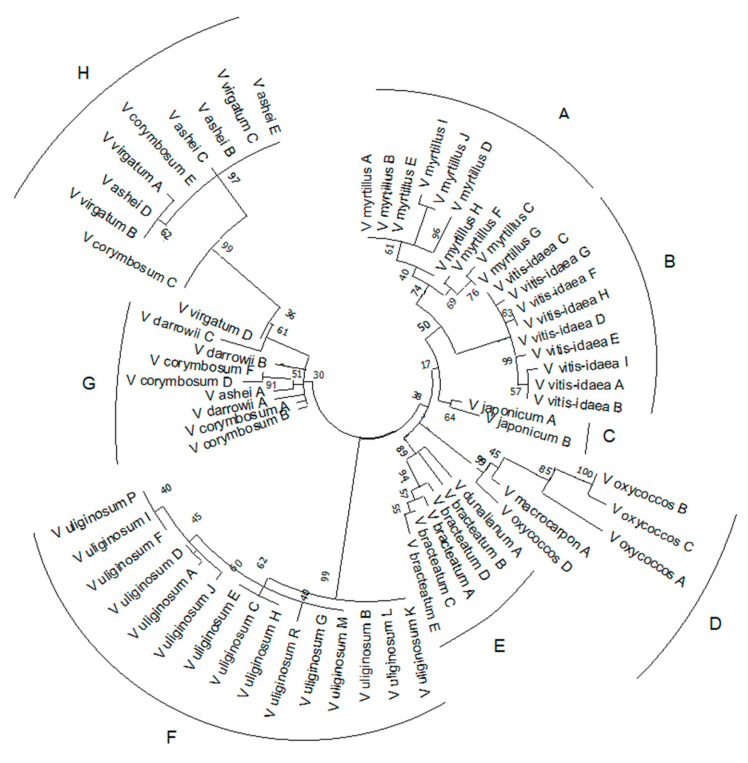
Phylogenetic relationships of alleles of *Vaccinium rolB/C*-like gene based on Maximum Likelihood method and General Time Reversible model (There were a total of 657 positions in the final dataset). (**A**) *V. myrtillus*, (**B**) *V. vitis-idaea*, (**C**) *V. japonicum*, (**D**) section *Oxycoccus*, (**E**) *V. bracteatum*, (**F**) *V. uliginosum*, (**G**,**H**) section *Cyanococcus*.

**Figure 5 ijms-24-06932-f005:**
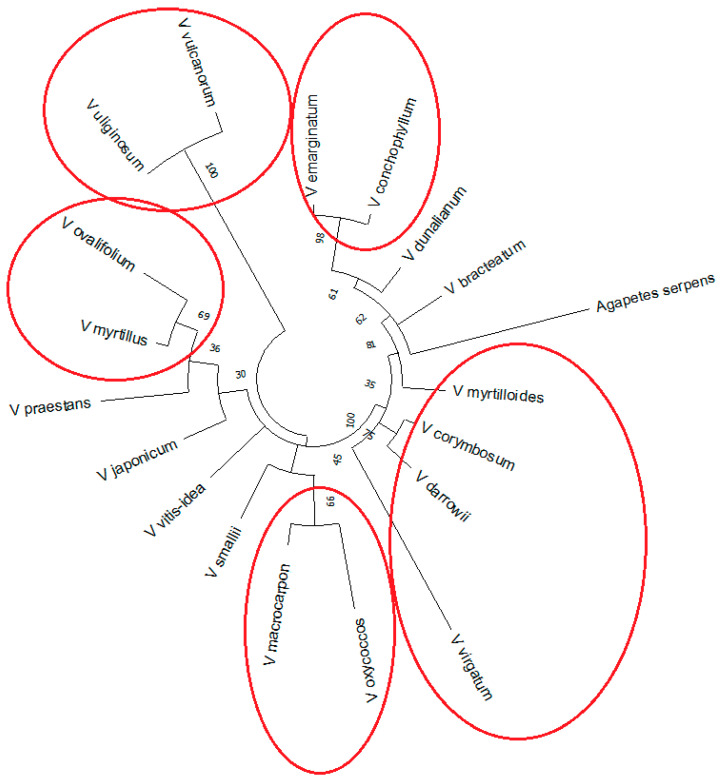
Molecular phylogenetic analysis of *Vaccinium* based on the *rolB/C*-like gene using Maximum Likelihood method and General Time Reversible model (There were a total of 667 positions in the final dataset). Representatives of the same section are circled in ovals.

**Figure 6 ijms-24-06932-f006:**
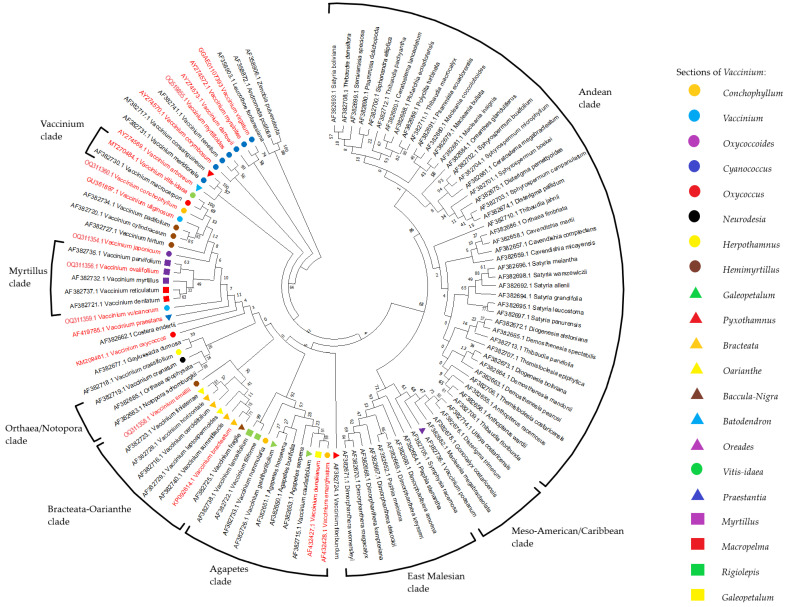
Molecular phylogenetic analysis of tribe Vaccinieae based on ITS sequences using Maximum Likelihood method and General Time Reversible model (There were a total of 651 positions in the final dataset). Species with sequences obtained by us are highlighted in red.

**Table 1 ijms-24-06932-t001:** Natural transgene of *Vaccinium* species, based on NGS data.

Section	Species	NGS Data	SRR Accession Number(s)
*Oxycoccoides* (Hooker f.) Sleumer	V. *japonicum* Miq.	*rolB/C*-like gene is present in the genome, and coverage is sufficient to assemble a full-length gene	SRR13349629
*Oxycoccus* (Hill) Koch	*V. macrocarpon* Aiton	*rolB/C*-like gene described earlier, in new SRA data coverage is sufficient to assemble a full-length gene	SRR13376387 SRR13376383 SRR13376388
	*V. oxycoccos* L.	*rolB/C*-like gene is present in the genome, and coverage is sufficient to assemble a full-length gene	SRR14876344
*Cyanococcus* A. Gray	*V. virgatum* Aiton/*V. ashei* Reade	*rolB/C*-like gene is present in the genome, and coverage is sufficient to assemble a full-length gene	SRR12686860 SRR12686864
	*V. darrowii* Camp	*rolB/C*-like gene is present in the genome, and coverage is sufficient to assemble a full-length gene	SRR13865261 SRR15673719
	*V. corymbosum* L.	*rolB/C*-like gene was described earlier, in new SRA data coverage is sufficient to assemble a full-length gene	SRR8298002 SRR8298006
	*V. stenophyllum* Steud.	*rolB/C*-like gene is present in the genome, but coverage is not sufficient to assemble a full-length gene	SRR19401505
	*V. tenellum* Aiton	*rolB/C*-like gene is present in the genome, but coverage is not sufficient to assemble a full-length gene	SRR19908712
*Hemimyrtillus* Sleumer	*V. smallii* A. Gray	*rolB/C*-like gene is present in the genome, but coverage is not sufficient to assemble a full-length gene	SRR19400793
*Myrtillus* Dumortier	*V. myrtillus* L.	*rolB/C*-like gene is present in the genome, and coverage is sufficient to assemble a full-length gene	SRR7523927 SRR7523930 SRR7523926
	*V. scoparium* Leiberg ex Coville	*rolB/C*-like gene is present in the genome, but coverage is not sufficient to assemble a full-length gene	SRR19405468
	*V. yatabe* Makino	*rolB/C*-like gene is present in the genome, but coverage is not sufficient to assemble a full-length gene	SRR19400792
*Polycodium* (Rafinesque) Rehder	*V. stamineum* L.	*rolB/C*-like gene is present in the genome, but coverage is not sufficient to assemble a full-length gene	SRR19401493
*Pseudocephalanthos* C.Y.Wu and R.C.Fang.	*V. dunalianum* Wight	*rolB/C*-like gene is present in the genome, and coverage is sufficient to assemble a full-length gene	SRR7768528 SRR2057022
*Bracteata* J.J. Smith	*V. bracteatum* Thunb.	*rolB/C*-like gene is present in the genome, and coverage is sufficient to assemble a full-length gene	SRR17459400 SRR17459403 SRR17459408
	*V. wrightii* A.Gray	*rolB/C*-like gene is present in the genome, but coverage is not sufficient to assemble a full-length gene	SRR19401672
*Vaccinium* L.	*V. uliginosum* L.	*rolB/C*-like gene is present in the genome, and coverage is sufficient to assemble a full-length gene	SRR7686610 SRR7686609
*Vitis-idaea* (Moench) Koch	*V. vitis-idaea* L.	*rolB/C*-like gene is present in the genome, and coverage is sufficient to assemble a full-length gene	SRR5799277 SRR5799278 SRR5799279
Relation to any section is not defined	*V. talamancense* (Wilbur and Luteyn) Luteyn	*rolB/C*-like gene is present in the genome, but coverage is not sufficient to assemble a full-length gene	SRR19401435
	*V. schoddei* Sleumer	*rolB/C*-like gene is present in the genome, but coverage is not sufficient to assemble a full-length gene	SRR19401400
	*V. stanleyi* Schweinf	*rolB/C*-like gene is present in the genome, and coverage is sufficient to assemble a full-length gene	SRR19354672
	*V. wollastonii* Wernham	*rolB/C*-like gene is present in the genome, but coverage is not sufficient to assemble a full-length gene	SRR19401324

**Table 2 ijms-24-06932-t002:** Studied genotypes of *Vaccinium* and their allelic status.

Species	Number of Genotypes	Number of Unique Alleles	Number of Homozygotes	Number of Heterozygotes	Ploidy of Studied Species	Max. Alleles per Plant
*V. myrtillus*	13	10	10	3	Diploids	2
*V. corymbosum*	2	6	0	2	Tetraploids	4
*V. macrocarpon*	3	1	3	0	Diploids	1
*V. oxycoccos*	4	4	3	1	Diploids, tetraploids and hexaploids	2
*V. virgatum/ashei*	5	9	3	2	Hexaploids	3
*V. japonicum*	2	2	2	0	Diploids	1
*V. vitis-idaea*	18	9	5	13	Diploids	2
*V. uliginosum*	14	18	0	14	Diploids, tetraploids and hexaploids	3
*V. dunalianum*	2	1	2	0	Diploids	1
*V. bracteatum*	3	5	1	2	Diploids	2
*V. darrowii*	2	3	0	2	Diploids	2

**Table 3 ijms-24-06932-t003:** Primers used in the work.

Primers		Tm	Amplicon Size	Purpose of Use
Name	Sequence			
VaccFvn	tcctaaccctaaccctgacc	50	1503	Amplification of the entire gene and flanking sequences
VaccRvn	aactcgtgattgtacctcgtt			
VaccR	ttactggccggtcctcatca	55	1002	Amplification and sequencing of the entire gene
VaccF	cacgtgtaaagcccgtgatgtt			
Vacc_ts_F	taaagcctgccactgcgatt	55	838	Efficient amplification of 838 out of 876 bp of coding sequence in any of studied *Vaccinium* genotype
Vacc_ts_pR	ccaatcgccacgagtaactaac			
Vacc_reR	ccgaagttcgccgtcctg	n/a *	n/a	Primers for sequencing of gene fragments
Vacc_reF	tccaagccatcgacctactc			
pJET1.2 Forward Sequencing Primer	cgactcactatagggagagcggc	58	n/a	Real-time PCR from colonies and sequencing
pJET1.2 Reverse Sequencing Primer	aagaacatcgattttccatggcag			

* n/a—not applicable.

## Data Availability

The sequences obtained during the work were deposited into the National Center for Biotechnology Information database (https://www.ncbi.nlm.nih.gov/ (accessed on 26 February 2023) with accession numbers OQ095269-OQ095321, OQ129379-OQ129399, OQ311354, OQ311356, OQ311358-OQ311360, and OQ519855.

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
