# Peer review of "Biodiversity of rolB/C-like Natural Transgene in the Genus Vaccinium L. and Its Application for Phylogenetic Studies"

_ijms, 2023, doi:10.3390/ijms24086932_

Round 1

Reviewer 1 Report

Minor Revision

In the manuscript titled “Biodiversity of rolB/C-like natural transgene in the genus Vac-2 cinium L. and its application for phylogenetic studies” authors provide insight into phylogenetic relationship among Vaccinium L. species based on their unique feature – the presence of agrobacterial rolB/C-like gene. An extensive investigation was performed without any significant mistakes in the design of the various analyses and the interpretation of the data. I suggest accepting this manuscript after it goes through minor revisions listed below.

Line 148. Why real-time PCR was used instead of conventional PCR for colony screening.

Line 238. The authors must provide amino acid sequences of all predicted proteins that were used for this analysis.

Line 256. Please indicate which primer pair was used for this analysis.

Line 257-258. It is not clear whether this numbering relates to gene CDS or amplicon length. Because VaccF primer is situated 121 bp upstream of the starting codon, this information needs more clarity.

Table 1. Include the exact PCR product lengths and Tm for each primer pair used. What does “Re-PCR” mean? Does it indicate real-time PCR or reamplification?

Figure 2. There is no "LO4" on the electrophoresis picture. Should it be "4"? Please include a scheme showing the exact positions of these mutations in the rolB/C-like gene.

Author Response

The authors thank the reviewer for careful reading of the manuscript, valuable comments and advice.
All corrections have been made to the text.

Line 148. Why real-time PCR was used instead of conventional PCR for colony screening.
** We have added clarification to the section "Materials and Methods"

Line 238. The authors must provide amino acid sequences of all predicted proteins that were used for this analysis.
**We have added Supplementary Table 3 and a link to it in the text

Line 256. Please indicate which primer pair was used for this analysis.

**We added primer names

Line 257-258. It is not clear whether this numbering relates to gene CDS or amplicon length. Because VaccF primer is located 121 bp upstream of the starting codon, this information needs more clarity.
**We have corrected the text.

Table 1. Include the exact PCR product lengths and Tm for each primer pair used. What does “Re-PCR” mean? Does it indicate real-time PCR or reamplification?
** We added additional columns to the table

Figure 2. There is no "LO4" on the electrophoresis picture. Should it be "4"? Please include a scheme showing the exact positions of these mutations in the rolB/C-like gene.
** We've made a correction to the figure's legend.

Meaningful changes are highlighted in yellow

Reviewer 2 Report

Dear authors!

The authors' article is of fundamental importance for phylogenetics. The obtained unique data on transgenes in representatives of Vaccinium L. will help to discover the mechanisms of genetic polymorphism of populations.

The authors described the occurrence of cT-DNA in different plant species. Scientists pointed out the difficulty of interpreting the results of phylogenetic analysis of Vaccinium L. when choosing other genome markers. The authors provided a convincing justification for the use of cT-DNA to revise the taxonomy of the taxon under study.

Molecular genetic and bioinformatic methods were used in the study.

The manuscript is beautifully illustrated, additional experimental material is presented. The results of the work are justified.

At the discretion of the authors, Figure 3 can be supplemented: add frequencies of alleles to the diagram. Although the authors have justified why this was not done. The addition of frequencies will allow a statistical comparison of the occurrence of alleles in Vaccinium species. If possible, I recommend conducting such an analysis.

Author Response

The authors thank the referee for careful reading of the manuscript, valuable comments and advice.

We added allele frequencies in diagrams of Figure 3

Since different species did not have the same alleles, there is no subject to comparison of their frequencies.

Reviewer 3 Report

Reviewer’s comments

The manuscript entitled “Biodiversity of rolB/C­-like natural transgene in the genus Vaccinium L. and its application for phylogenetic studies” is of great interest for plant scientist community especially working in berries breeding. Although the manuscript contains good information but there are some flaws that must be addressed and fulfilled for the validation of study and outputs. For improvement of manuscript, consider the following suggestions and comments.

Introduction: English language standard of the this section is quite poor. Few of the mistakes has been highlighted in the attached file. Many of the sentences are too long, which makes them difficult to understand. Although this section provides a detailed history of previous work done on characterization and phylogenetic studies conducted in genus Vaccinium  and its related genra, but no clear objectives are given at the end of introduction section. Please state a clear objective /hypothesis that why this study was conducted, and what could be the benefits of the outcomes reported.

Materials and Methods: Give details about DNA extraction protocol.

Discussion: This is the weak section of the paper, poorly writer, and supported by very few references. More and recent citations needs to be added, and the language of this section (specifically) needs to be improved. At many points, authors used lengthy sentences, which make it difficult for the reader(s) to understands. Also noted, repetition of arguments without any logical justification.

General: English language needs considerable improvement. Many spellings and grammatical mistakes has been highlighted in the attached file also.

Conclusion: Conclusion is too brief, please elaborate it with addition of few more sentences and clarify the findings of the research.

Decision: Paper may be accepted (with minor revisions) after incorporation of suggestions/comments by the authors.

Author Response

We would like to express our gratitude to the reviewer for the usefull comments
In accordance with these comments, we revised our manuscript.

Specifically,  the manuscript was checked by our native English-speaking colleague.
In the introduction,  objective of our study is formulated and future applications of the outcomes reported.

In the Material and Methods section we have specified the DNA extraction method. Since this method is widely known, we did not describe its details, but limited ourselves to a reference to the protocol.
More citations have been added to the discussion to substantiate our statements. 
The conclusion is expanded  to clarify the findings of the research.

Substantial changes highlighted in yellow